# Agility and Innovativeness: The Serial Mediating Role of Helping Behavior and Knowledge Sharing and Moderating Role of Customer Orientation

**DOI:** 10.3390/bs12080274

**Published:** 2022-08-08

**Authors:** Sungjin Park, Keuntae Cho

**Affiliations:** 1Graduate School of Management of Technology, Sungkyunkwan University, Seoburo 2066, Suwon 16419, Korea; 2Department of Systems Management Engineering, Graduate School of Management of Technology, Sungkyunkwan University, Seoburo 2066, Suwon 16419, Korea

**Keywords:** agile, agility, innovativeness, helping behavior, knowledge sharing, customer orientation

## Abstract

This study aims to understand the mechanism whereby the Agile approach works by analyzing the effect of agility on innovativeness, the sequential mediating effect of helping behavior and knowledge sharing, and the moderating effect of customer orientation. Data for 323 Information and Communication Technology (ICT) companies and 964 non-ICT companies were collected and analyzed through online surveys. Bootstrapping analysis using Model No. 83 of the PROCESS macro confirmed that agility increases team members’ helping behaviors and strengthens knowledge sharing, which in turn has a positive effect on innovativeness. More specifically, helping behavior and knowledge sharing sequentially mediate the relationship between agility and innovativeness. In addition, the study verified that customer orientation moderates the effect of agility on helping behavior. This study is meaningful in showing that it is important to create a culture that pursues “customer value” while promoting mutually helping behavior and sharing knowledge when introducing Agile methodology.

## 1. Introduction

Owing to the impact of COVID-19, which has limited face-to-face situations, along with the development of sensors and GPU (Graphics Processing Unit), big data, and artificial intelligence technologies, many industries are rapidly pursuing digital transformation for survival. In addition, interbreeding has appeared between different categories of technology and the boundaries between industries have blurred, increasing volatility, uncertainty, complexity, and ambiguity in the business environment. Therefore, a variety of strategies and methodologies have emerged that can respond more effectively to changes in the environment.

A prime example of this is Agile. Agile development methodology, which has been proposed as a software development methodology, is the opposite of the existing waterfall model. Unlike the waterfall model, in which a perfect plan is established in advance and implemented sequentially, Agile methodology plans gradually on the premise of high uncertainty and implements it quickly, examining customer and market responses. By constantly iterating these small experiments and pivoting their plans, companies can overcome uncertainty by enhancing organizational agility for survival and minimizing the risk of failure. In addition, the Agile approach is being introduced and spread not only in software development companies but also in non-software development fields, including the financial industry [1,2].

Therefore, several studies of Agile methodology are ongoing. However, Agile methodology is not based on academic theories, but has arisen from the design, practice, and structure of various technologies and tools in the software development industry. Most previous studies have focused on the effectiveness of Agile practices in the software development field rather than in the fundamental work mechanism of Agile [3]. Not only in research but also in actual business, although the term “Agile” has been successfully popularized, the number of agile organizations did not increase because they focused on imitation by introducing only formal Agile practice techniques and tools [4]. Therefore, to properly apply the Agile approach in all industries, including software and non-software development, attention should be placed on agility rather than the effectiveness of fragmented Agile practices and tools [5].

Agility refers to an organization’s ability to respond effectively and quickly to changes in the market, supply, and demand in the development of competitive behavior and opportunities for innovation [6,7]. Agility is also known to have a positive effect on organizational performance, including innovativeness [8,9]. However, parts of previous studies of agility remains unexplained.

First, there is a dearth of research on the mechanism whereby agility affects innovation. To the extent that mediation and moderation between independent and dependent variables explain such a mechanism, such research can determine the essential reason that agility affects innovativeness through the mediation and moderation between agility and innovativeness. In particular, because Agile is a process in which work is performed by sets of teams rather than individuals, the work mechanism between agility and innovativeness can be modelled with the input-process-output (IPO) model proposed by Hackman [10]. The IPO model is widely accepted in team performance [11,12] and is also used theoretically in studying innovation and creativity [13]. Therefore, it is necessary to study the factors that shape the team process between agility, which is the input factor of teams, and innovation, which is their output factor. This study tests the hypothesis that agility increases innovativeness by sequentially mediating helping behaviors and knowledge sharing.

Second, it is theoretically and empirically valid to state that agility affects innovation [8,9]. However, it is too simplistic to claim that such a phenomenon is the same in all situations. This is because the relationship between agility and innovativeness may manifest differently depending on the situation or context. In other words, it is possible that moderating variables exist between these two variables. Identifying any moderator variables would allow the relationship between the two variables to be explained more precisely; therefore, research on such moderator variables is necessary. This study aims to establish and test the hypothesis that customer orientation moderates the relationship between agility and helping behavior.

Agility can improve helping behaviors. Helping behaviors benefit others, and thus the organization to which they belong [14]. When responding to changes in the environment, employees often need to perform work urgently beyond their respective boundaries, which can naturally increase their helping behaviors [15]. These helping behaviors bring positive results to people and organizations receiving help by sharing resources, providing ideas for problem-solving, or providing direct work assistance [16,17,18,19], whereby agility can increase organizational innovativeness [20,21,22]. Therefore, this study examines the mediating effect of agility on innovativeness through the sequential mediation of helping behaviors and knowledge sharing.

In addition, this study suggests that customer orientation moderates the relationship between agility and helping behaviors. Customer orientation refers to an attitude toward customers that seeks to satisfy their needs from their viewpoint as much as possible [23]. Previous studies have highlighted the importance of “integration” as a variable, moderating the process by which agility influences helping behavior [24,25,26]. This is because employees respond to agility differently according to their own standards, which can lead to confusion. Therefore, integration must be achieved for employees to make consistent decisions, and Agile development methodology emphasizes that this integration must be based on the customer orientation in particular [27]. More specifically, when customer orientation is high, the effect of agility on helping behavior can be strengthened; and the effect can be reduced when customer orientation is low. Therefore, this study examines the moderating effect of customer orientation on the relationship between agility and helping behavior.

In summary, this study aims to determine whether agility affects innovation through the sequential mediation of helping behaviors and knowledge sharing. Additionally, this study examines whether customer orientation reinforces the positive relationship between agility and helping behaviors.

## 2. Theoretical Background and Hypotheses

### 2.1. Agile Development Methodology

Agile development methodology is a project management methodology that originated in the field of software development. The early model of project management for software development originally borrowed the waterfall model, following the project management methodology of architecture and civil engineering. The waterfall model takes a long time to completely analyze customer requirements and design a specific plan according to which results are developed and delivered. However, because of the nature of software, unlike in architecture and civil engineering, the results can be modified and supplemented relatively easily. Therefore, even when a customer’s requirement analysis and plan design are conducted over a long time, the customer’s requirements frequently change and increase over time while the software is developed according to the plan. Therefore, project management methods for software development have evolved into Agile development models instead of waterfall models. The Agile model is designed to prioritize key customer needs, develop them quickly, deliver them to customers, receive feedback, address additional needs within shorter cycles, and iterate the incremental improvements. Figure 1 shows the differences between the waterfall and Agile models [28].

In this context, Agile development methodologies, such as SCRUM and XP, have been proposed since the 1990s, and some software developers gathered in 2001 to issue the Agile Manifesto [29] to organize the direction of the increasing number of Agile development methodologies. Accordingly, the Agile Manifesto presents the core values and philosophy for designing and developing software rather than prescribing specific techniques. The four core values of the Agile Manifesto are (1) individuals and interactions over processes and tools, (2) working software over comprehensive documentation, (3) customer collaboration over contract negotiation, and (4) responding to change over following a plan.

Agile development methodologies newly born after the announcement of the Agile Manifesto and those created before it have been reorganized and improved in these directions. This Agile development methodology has resulted in many achievements in software. In a situation where the recent business environment is rapidly becoming uncertain, the Agile model has received attention not only from software development companies but also from non-software development fields, including the financial industry [1,2].

Accordingly, several studies have been conducted on the effects of Agile. However, as Agile methodology was not based on academic theories but devised, practiced, and structured based on various practices and tools in the software development industry, most previous studies have focused on the fragmentary effects of Agile practices and tools in software development [3].

However, adopting Agile in the workplace is not an easy task. While some organizations have been successful in adopting Agile, many have failed. According to statistics, 47% of the organizations adopting Agile have failed [30]. Not only statistically but in practice in business, there are voices of self-reflection that the term ”Agile” has become a vogue, but agile organizations are actually not increasing, and this is because they rush introducing formal Agile practices and tools [4].

Therefore, to properly utilize Agile in all industries, including Information and Communication Technology (ICT), it is more important to understand the work mechanism of Agile than the effect of superficial Agile practices and tools. However, because the concepts and constructs of Agile are not yet academically organized and widely used, this study generalizes the core value of the Agile Manifesto [29], which is the official beginning of Agile, to a similar existing academic construct, as shown in Table 1, examines the work mechanism of Agile, and establishes and verifies related hypotheses.

### 2.2. Agility and Innovativeness

Agility is an organization’s ability to respond effectively and quickly to changes in the market, supply, and demand in the development of opportunities for competitive action and innovation [6,7]. It has a positive impact on organizational performance [8,9]. Innovativeness is one of the most representative types of organizational effectiveness and refers to an organization’s openness to new ideas and experimental processes [31]. In responding quickly to environmental changes, organizations naturally become open to the application of new ideas or experimental processes. Therefore, the following hypothesis is established:

**Hypothesis** **1.**
*Agility positively affects innovativeness in ICT companies.*


### 2.3. Serial Mediating Role of Helping Behavior and Knowledge Sharing

According to the IPO model proposed by Hackman [10], agility, an organizational and environmental input factor, produces innovativeness as a team output through the team process. This study suggests that helping behaviors and knowledge sharing as a team process sequentially mediate agility and innovativeness. In other words, agility can increase the level of helping behaviors of team members, which increases knowledge sharing among them. Eventually, this enhanced knowledge sharing increases the level of innovativeness.

#### 2.3.1. Agility and Helping Behavior

Helping behaviors benefit other team members and their organization [14]. Helping behaviors refer to the degree of interaction among employees who depend on and cooperate with each other to perform tasks efficiently [32,33,34]. In particular, when employees respond quickly to severe environmental changes, they often perform tasks urgently beyond their individual roles, which naturally increases their helping behaviors [15]. Therefore, the following hypothesis is established:

**Hypothesis** **2.**
*Agility positively affects helping behaviors in ICT companies.*


#### 2.3.2. Helping Behavior and Knowledge Sharing

Helping behavior brings positive results to individuals and organizations receiving help, such as sharing resources, providing ideas for problem-solving, and providing direct work assistance [16,17,18,19]. On the other hand, knowledge sharing refers to the degree to which team members share work-related ideas, information, and suggestions with each other [35]. Sharing ideas, information, and suggestions for problem-solving with team members is part of this knowledge-sharing activity and helping behavior. Knowledge sharing encompasses the sharing of knowledge related to overall work, as well as knowledge that is directly helpful in solving problems. Therefore, the following hypothesis is established:

**Hypothesis** **3.**
*Helping behavior positively affects knowledge sharing in ICT companies.*


#### 2.3.3. Knowledge Sharing and Innovativeness

Knowledge sharing has become the basis for new knowledge creation [36]. The fragments of different knowledge that each individual possesses are shared, interact with each other, and are constructed and recreated as new knowledge [37]. Additionally, this expanded knowledge promotes the generation of creative ideas by various members [38]. Therefore, the following hypothesis is established:

**Hypothesis** **4.**
*Knowledge sharing positively affects knowledge innovativeness in ICT companies.*


#### 2.3.4. Serial Mediating Role of Helping Behavior and Knowledge Sharing

As previously mentioned, agility improves the level of helping behavior in the work performed by team members, and increased helping behavior increases their knowledge sharing, which eventually increases the level of innovativeness. Accordingly, the following hypothesis for the serial mediating effect is proposed:

**Hypothesis** **5.**
*Helping behavior and knowledge sharing sequentially mediate the relationship between agility and innovativeness in ICT companies.*


### 2.4. Moderating Role of Customer Orientation

Theoretically and empirically, agility affects innovation [8,9]. However, it is an overly simplistic claim that the same phenomenon occurs for all members under all circumstances. This is because the relationship between agility and innovativeness may manifest differently in different situations or contexts. In other words, it is possible that a moderating variable exists between agility and innovativeness. Whose identification would allow the relationship between the two variables to be explained more precisely; therefore, studies exploring these moderator variables are necessary.

On the other hand, this moderating effect can be explained by Martin and Cullen’s [39] context–attitude–behavior model. According to this view, various social and environmental contexts within an organization act as decisive factors in shaping the attitudes and behaviors of members within the organization [39,40]. Agility in a rapidly changing environment, which functions as an important social context surrounding the team, can influence their helping behaviors by moderating customer orientation and the attitude of members focusing on customers to respond quickly to the environment (context: agility–attitude; customer orientation–behavior: helping behavior). Customer orientation refers to seeking to fully understand the target customers and continue to create superior value for them [41]. Accordingly, the following hypothesis for the serial mediating effect is proposed:

**Hypothesis** **6.**
*Customer orientation moderates the relationship between agility and helping behavior in ICT companies.*


Summarizing these hypotheses, agility enhances the helping behavior of organizational members, thereby strengthening knowledge sharing and eventually improving innovativeness. Additionally, the influence of agility on helping behavior varies according to the level of customer orientation. To empirically verify this, a research model, as shown in Figure 2, is constructed by setting agility as an independent variable, innovativeness as a dependent variable, helping behavior and knowledge sharing as sequential mediating variables, and customer orientation as a moderating variable.

## 3. Methodology

### 3.1. Sample and Procedure

The data were collected through an online survey. First, as Agile development methodology started in the software development, and in order to examine the work mechanism of Agile in ICT companies, data were collected from Korean ICT companies, with a target goal of 300 respondents in May 2021. Finally, data of 323 respondents working in eight Korean ICT companies were collected and analyzed.

In addition to the ICT sector, many non-ICT companies are interested in and introduce Agile methodology; therefore, in order to examine the work mechanism of Agile in non-ICT companies, data were also collected from Korean non-ICT companies, with a target goal of 1000 respondents in June 2021. This larger target was set because it was considered that much more data were needed than from ICT companies to analyze the effectiveness of the work mechanism of Agile because the Agile development methodology started with software development. Finally, data of 964 respondents working in 29 Korean non-ICT companies were collected with the help of members of a community of practitioners involved in human resources.

Table 2 shows that respondents’ age, educational background, number of years worked, and position were collected evenly, similar to the normal distribution.

### 3.2. Measures

In this study, agility was set as an independent variable, helping behavior and knowledge sharing as mediating variables, customer orientation as a moderating variable, and innovativeness as a dependent variable. All variables were measured using a 5-point Likert scale (1 = strongly disagree, 5 = strongly agree) at the team level.

#### 3.2.1. Agility

Three operational agility measurement items were adopted from Sambamurthy et al. [6] to measure agility: “My team has the ability to appropriately scale up or down the level of work to meet environmental variability”, “My team has the ability to quickly respond to mistakes or obstacles that may be caused by partner companies or related teams”, and “My team has the ability to respond quickly to changes in customers (or stakeholders)”.

#### 3.2.2. Helping Behavior

Four items were adopted from Fleishman [42] to measure helping behavior: “My team members are willing to take their time to help other team members with work-related problems”, “My team members show genuine interest and courtesy to each other even in difficult situations”, “My team members make time to help other team members who are struggling with individual problems”, and “My team members voluntarily support the work of other team members”.

#### 3.2.3. Knowledge Sharing

To measure knowledge sharing, we adopted four items from Srivastava et al. [35]: “My team members share their knowledge and know-how with colleagues”, “My team members are willing to teach coworkers what they know how to do”, “My team members exchange and share their information, knowledge, and skills with colleagues”, and “My team members provide hard-to-find knowledge or expertise to their colleagues without conditions”.

#### 3.2.4. Customer Orientation

To measure customer orientation, we adopted three items from Saxe and Weitz [43]: “My team answers customer (or stakeholder) questions as accurately as possible”, “My team strives to quickly resolve complaints from customers (or stakeholders)”, and “My team communicates well with customers (or stakeholders) to understand their needs”.

#### 3.2.5. Innovativeness

To measure innovativeness, we adopted three items from Covin and Slevin [31]: “The way my team operates is creative”, “My team is often the first to try new features and methods”, and “My team actively adopts innovative management practices and ways of working”.

Table 3 summarizes the variables used in this study.

### 3.3. Analytical Approach

In this study, SPSS 26.0 and AMOS 26.0 programs were used to analyze the collected data. Exploratory and confirmatory factor analyses were performed to confirm the validity of each item of the variables, and the reliability of each variable was measured using Cronbach’s α. Frequency analysis was conducted to examine the characteristics of the respondents, such as sex, age, educational background, number of years worked, and position.

Bootstrapping was performed using Model No. 83 of the PROCESS macro proposed by Hayes [44] to verify the serial mediating effect of helping behavior and knowledge sharing in the relationship between agility and innovativeness (agility → helping behavior → knowledge sharing → innovativeness) and moderating effect of customer orientation in the relationship between agility and helping behavior (agility → helping behavior). To obviate the problem of multicollinearity of the interaction term between the independent and moderating variables, the variables were centered on the mean and used for analysis.

## 4. Results

### 4.1. Validity and Reliability

Exploratory factor analysis was performed to confirm the validity by integrating ICT and non-ICT data, and Cronbach’s α was used to confirm reliability. As shown in Table 4, both the validity and reliability of the variables are confirmed to be appropriate.

Confirmatory factor analysis was performed for additional validity analysis. As shown in Table 5, RMR (<0.05), GFI (>0.9), AFGI (>0.9), CFI (>0.9), TLI (>0.9), NFI (>0.9), IFI (>0.9), and RMSEA (<0.05) were all suitable.

### 4.2. Hypotheses 1–4 (Direct Effect) Testing

Table 6 and Table 7 show the values of the coefficients for each variable path in the data analysis results for ICT and non-ICT companies, respectively. The effects of each pathway on the ICT and non-ICT data were all statistically significant.

First, in the results of data analysis for ICT companies, agility had a positive effect on innovativeness (β = 0.501, *p* < 0.001) and helping behavior (β = 0.346, *p* < 0.001). Helping behavior had a positive effect on knowledge sharing (β = 0.665, *p* < 0.001), and knowledge sharing had a positive effect on innovativeness (β = 0.179, *p* < 0.001).

Second, in the results of the data analysis for non-ICT companies, agility had a positive effect on innovativeness (β = 0.486, *p* < 0.001) and helping behavior (β = 0.322, *p* < 0.001). Helping behavior had a positive effect on knowledge sharing (β = 0.598, *p* < *0*.001), and knowledge sharing had a positive effect on innovativeness (β = 0.169, *p* < 0.001).

As a result, Hypotheses 1, 2, 3, and 4 are supported in ICT and even in non-ICT categories.

### 4.3. Hypothesis 5 (Serial Mediating Effect) Testing

For helping behavior and knowledge sharing to be set as mediating variables, the effect of agility as an independent variable on innovativeness as a dependent variable and on helping behavior and knowledge sharing should be statistically significant. In addition, when mediating variables, such as helping behavior and knowledge sharing, are added, the effect of agility as an independent variable on innovativeness as a dependent variable should be reduced [45,46]. In the data analysis results of this study, as shown in Table 6 and Table 7, the effect between variables was statistically significant.

In addition, as a result of the data analysis for ICT companies, the direct effect (β = 0.501, *p* < 0.001) of agility on innovativeness when helping behavior and knowledge sharing were added as mediating variables rather than the overall effect (β = 0.624, *p* < 0.001) of agility on innovativeness decreased, indicating that helping behavior and knowledge sharing mediated the relationship between agility and innovativeness. In addition, data analysis for non-ICT companies showed that the direct effect (β = 0.486, *p* < 0.001) of agility on innovativeness when helping behavior and knowledge sharing were added as mediating variables rather than the overall effect (β = 0.620, *p* < 0.001) of agility on innovativeness decreased, indicating that helping behavior and knowledge sharing mediated the relationship between agility and innovativeness. In other words, helping behavior and knowledge sharing mediated the relationship between agility and innovativeness in ICT and non-ICT categories.

To determine whether there is a sequential double mediation effect of helping behavior and knowledge sharing in the relationship between agility and innovation, 5000-time bootstrapping was performed with the confidence interval set to 99% to analyze the effect. The results of the effect verification are presented in Table 8.

As a result of verifying the sequential double mediating effect of helping behavior and knowledge sharing between agility and innovativeness (agility → helping behavior → knowledge sharing → innovativeness), the 99% confidence interval for ICT companies was 0.002–0.098 (β = 0.041, *p* < 0.01), in which a value of 0 was not included, indicating that the sequential double mediation effect was statistically significant. The 99% confidence interval for non-ICT companies ranged from 0.012 to 0.058 (β = 0.033, *p* < 0.01), which did not include the value of 0, indicating that the sequential double mediation effect was statistically significant. Thus, Hypothesis 5 is supported in both ICT and non-ICT companies.

### 4.4. Hypothesis 6 (Moderating Effect) Testing

Table 9 shows the interaction term of agility as an independent variable and customer orientation as a moderating variable, which has no significant effect (β = 0.047, *p* = 0.176) on helping behavior in the data analysis for ICT companies. In other words, this finding does not support Hypothesis 6 in the ICT field.

However, Table 10 shows the interaction term of agility and customer orientation, which has a significant effect (β = 0.063, *p* < 0.01) on helping behavior in the data analysis results for non-ICT companies. In other words, in the non-ICT sector, agility positively affects innovativeness through the sequential double mediation of helping behavior and knowledge sharing (agility → helping behavior → knowledge sharing → innovativeness), and customer orientation moderates the relationship between agility and helping behavior (agility → helping behavior).

Although Agile development methodology first started in the ICT, the reason customer orientation had no effect in moderating the relationship between agility and helping behavior is that customer orientation acted as an antecedent variable that directly affects helping behavior, not as a moderating variable. As shown in Table 9 of the data analysis for ICT companies, the effect of customer orientation on helping behavior (β = 0.323, *p* < 0.001) is as significant as that of the independent variable agility on helping behavior (β = 0.346, *p* < 0.001). Table 10 shows that even in the data analysis for non-ICT companies, the direct effect of customer orientation on helping behavior (β = 0.279, *p* < 0.001) is significant. This is because owing to the characteristics of the industry in the ICT sector, product/service conversion is relatively easy; therefore, they are already sensitive to customer needs and agile responses to changes in the business environment. Therefore, customer orientation has a huge direct effect on helping behavior, meaning that the moderating effect of customer orientation on the relationship between agility and helping behavior is relatively insignificant. Therefore, it is possible to modify the research model to show that customer orientation not only moderates the relationship between agility and helping behavior but also directly affects helping behavior.

On the other hand, as previously mentioned, the moderating effect of customer orientation on the relationship between agility and helping behavior is significant in the non-ICT sector. Table 11 presents the results of analyzing the conditional effect by dividing customer orientation into high (+1SD) and low (−1SD) groups, showing that customer orientation values were β = 0.385, *p* < 0.01 (+1SD) and β = 0.260, *p* < 0.01 (−1SD) in the high and low groups, respectively, and were both significant. In other words, the effect of agility on helping behavior was significant in the area where customer orientation value was ±1SD.

In the overall research model, in which agility affects innovativeness through sequential double mediating helping behavior and knowledge sharing (agility → helping behavior → knowledge sharing → innovativeness), as shown in Table 12, the results for the conditional effect showed that customer orientation was β = 0.026, *p* < 0.01 and β = 0.039, *p* < 0.01 in the low (−1SD) and high (+1SD) groups, respectively, and were both significant. In other words, the effect of agility on innovativeness, through the serial mediation of helping behavior and knowledge sharing, was also significant in the area where the customer orientation value was ±1SD.

To more easily understand the moderating effect of customer orientation on the relationship between agility and helping behavior, it was visualized by dividing customer orientation into high (+1SD) and low groups (−1SD). Figure 3 shows that the higher the customer orientation, the higher the rate of increase in helping behavior with the increase in agility.

## 5. Discussion and Implications

To properly use Agile methodology in all industries, including ICT and non-ICT, understanding how Agile works (Agile work mechanism) is more important than adopting superficial Agile practices and tools. However, as Agile methodology is not based on academic theories, it arises from the design, practice, and structure of various techniques and tools in the software development industry. Most previous studies have focused on the effectiveness of Agile practices in software development rather than its fundamental work mechanism [3,47]. As a result, the concepts and constructs of Agile have not yet been academically organized and widely used [48]. Therefore, in this study, the core values of the Agile Manifesto [29], which are the reference point of Agile, were generalized to similar existing academic concepts such as agility, customer orientation, helping behavior, and knowledge sharing, and then a research model was established with constructs and verified to understand the Agile work mechanism.

This study confirmed that agility increases the level of helping behavior and knowledge sharing among members, which in turn improves their innovativeness (agility → helping behavior → knowledge sharing → innovativeness), as a work mechanism of Agile. In particular, data were collected and analyzed not only for ICT companies, which are considered particularly suitable for adopting Agile methodology, but also for non-ICT companies that have recently become interested in Agile and are attempting to introduce it.

In the process, it was confirmed that customer orientation reinforces the relationship between agility and helping behavior in non-ICT companies and that the direct effect of customer orientation on helping behavior is significant. However, the moderating effect of customer orientation on agility and helping behavior was not significant in ICT companies; rather, the direct effect of customer orientation on helping behavior was strengthened as much as the effect of the independent variable agility on helping behavior. Therefore, it is not enough to say that the moderating effect is significant in non-ICT companies and not significant in ICT companies. It can be inferred that in industries in which product/service conversion is relatively easy, customer orientation has a direct effect on helping behavior along with agility, whereas in industries in which product/service conversion is relatively difficult, customer orientation indirectly reinforces the relationship between agility and helping behavior. Therefore, customer orientation, which is an independent variable, is an important antecedent factor.

### 5.1. Theoretical Implications

This study focused on elucidating the working mechanism of Agile methodology. To respond quickly to the rapidly changing business environment, ICT and non-ICT companies have turned to Agile methodology, which has thus become a major topic of research [1,2,47]. However, Agile methodology is not based on academic theories but arises from the design, practice, and structure of various techniques and tools in the software development industry. Most previous studies have focused on the effectiveness of Agile practices rather than its fundamental work mechanism [3,47]. However, this study established a research model with constructs and verified it to understand the Agile work mechanism. This contributes to expanding the scope of interest in the study of Agile, focusing on the consequent effectiveness of Agile and its fundamental work mechanism.

Second, it contributed to refining how agility leads to organizational innovation. Previous studies have found that agility has a positive effect on organizational performance, including innovativeness [8,9,49]. However, parts of previous studies of agility have not been fully explained, and in certain respects, these previous studies do not fully explain how agility increases organizational innovation. Therefore, this study verified that agility increases organizational innovation through helping behavior and knowledge sharing. When employees respond quickly to severe environmental changes, they often perform tasks urgently beyond their individual role, which naturally increases their helping behavior. Subsequently, helping behavior increases employee’s knowledge sharing and eventually increases their level of innovativeness.

Finally, in the process of agility influencing organizational members’ behaviors (helping behavior and knowledge sharing), by revealing that the organization’s purpose and members’ consensus (customer orientation) act as moderators, the scope of agility studies has expanded. A quick response means that the direction of the response is flexible. When the direction of the organization and consensus of the members are added, the power of agility is strengthened by aligning them without being dispersed. From this perspective, it is important to identify the moderating factors of an organization that either reinforce or weaken the impact of agility on the effectiveness of the organization.

### 5.2. Practical Implications

In introducing Agile methodology, rather than rushing to adopt formal and superficial practices and tools, the focus should be on creating a culture in which organizational members can open up transparently about each other’s work, actively share knowledge and ideas, and voluntarily help each other. When members of an organization are not interested in other people’s work because they are focused on their own tasks and their work and knowledge are not shared, the effectiveness of Agile, despite practicing Agile methodologies and tools, is greatly reduced. This is also related to why 47% of the organizations that adopt Agile methodology were found to have failed in a statistical study [30].

Second, when the orientation (purpose and vision) of the organization and consensus of the members are added, the power of agility is strengthened without being scattered, as mentioned earlier. Agility is based on autonomy; therefore, when the direction is not aligned, the overall efficiency and effectiveness of the organization is reduced. However, it is difficult to clarify the direction of the organization when changes in the business environment are severe. In this situation, customer orientation can be good for an organization. When the organization’s orientation is focused on creating customer value and consensus is formed with the members of the organization, individual agility can have autonomy and, at the same time, be aligned with priorities of decision-making.

Finally, the Agile approach can be utilized not only in ICT companies but also in non-ICT companies by understanding and implementing the work mechanism of Agile. This is because even though companies do not formally adopt a specific Agile methodology, they can implement and benefit from their own Agile approach in a way that promotes organizational agility, customer orientation, helping behavior, and knowledge sharing.

## 6. Conclusions

This study aims to understand the working mechanism of the Agile approach by analyzing the effect of agility on innovativeness, the sequential mediating effect of helping behavior and knowledge sharing, and the moderating effect of customer orientation. In particular, these research hypotheses are supported not only in ICT companies, which are considered suitable for Agile methodologies, but also in non-ICT companies. Specifically, it was found that helping behavior and knowledge sharing sequentially mediate the relationship between agility and innovativeness in both ICT and non-ICT companies. However, it was only partially supported that customer orientation moderates the relationship between agility and helping behavior. In non-ICT companies, customer orientation was confirmed to strengthen the relationship between agility and helping behavior, but it was not significant in ICT companies. On the other hand, customer orientation was meaningful in that it directly affects helping behavior in both ICT and non-ICT companies. This can be interpreted in two ways in future research. First, it can be inferred that customer orientation indirectly reinforces customer orientation in industries where product/service conversion is relatively easy, whereas customer orientation directly affects helping behavior along with agility in industries where product/service conversion is relatively difficult. Second, follow-up research can be conducted by setting customer orientation as an independent variable equal to agility in its direct effects on helping behavior.

This study is meaningful because most previous studies have focused on the effectiveness of Agile practices in the software development field rather than on the fundamental work mechanism of Agile [3,47]. This study focuses on understanding the working mechanism of Agile approach and suggests practical implications that it is important to create a culture that pursues “customer value” while promoting mutually helping behavior and sharing knowledge when introducing Agile methodology based on understanding the work mechanism of Agile approach. Not only in research but in business practice, although the term “Agile” succeeded in popularizing the term, many organizations have failed to adopt Agile [30], and so this study should be useful for companies adopting Agile.

Although this study has practical implications, it also has certain limitations. First, it is based on a survey of individual members’ perceptions of team-level variables. In future studies, it would be useful to measure team indicators more precisely. Second, as the study and definition of the academic construct of Agile are lacking, this study borrowed similar constructs such as agility, customer orientation, helping behavior, and knowledge sharing. When the academic concepts and constructs for Agile methodology are defined and measured, it is possible to examine the work mechanism of Agile more precisely. Finally, this study focuses on the core values of Agile and factors in the early stages. The other characteristics and commonalities of Agile methodologies are short-cycle iterations, reflections, retrospectives, adjustments and refinements of original plans, and incremental improvements. Agile innovation is not achieved instantly but through several iterations and improvements; therefore, it is suggested that these variables, which were not covered in this study, be included in future studies.

## Figures and Tables

**Figure 1 behavsci-12-00274-f001:**
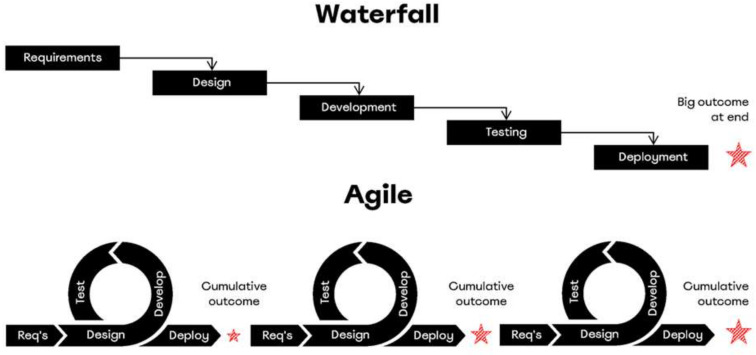
Waterfall and Agile models.

**Figure 2 behavsci-12-00274-f002:**
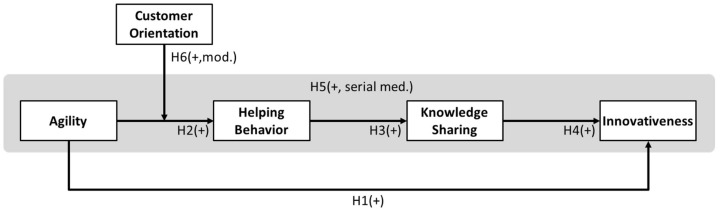
The working mechanism research model of the Agile approach.

**Figure 3 behavsci-12-00274-f003:**
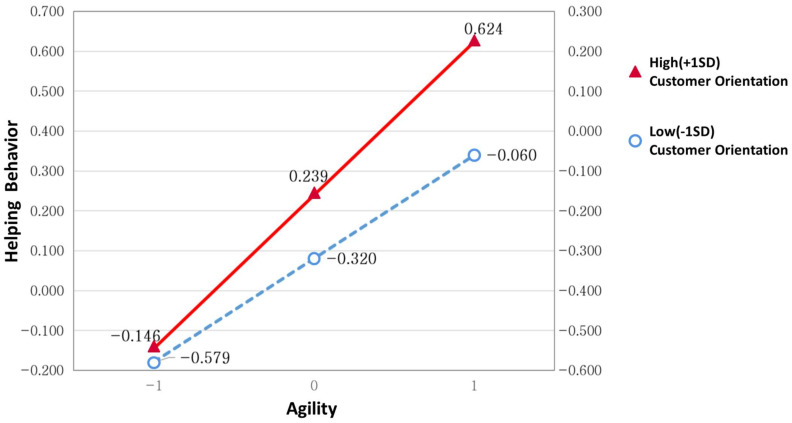
Moderating effect of customer orientation.

**Table 1 behavsci-12-00274-t001:** Agile core values and similar concepts and constructs.

Core Values of the Agile Manifesto	Construct
• Responding to change over following a plan	Agility
• Individuals and interactions over processes and tools	Helping behavior, Knowledge sharing
• Customer collaboration over contract negotiation	
• Working software (deliverables) over comprehensive documentation	Customer orientation

**Table 2 behavsci-12-00274-t002:** Respondents’ descriptive statistics.

		ICT Companies	Non-ICT Companies
Age	20s	2.5% (8)	7.8% (75)
	30s	16.4% (53)	29.1% (281)
	40s	42.4% (137)	40.7% (392)
	50s	38.7% (125)	21.8% (210)
	60s	0.0% (0)	0.6% (6)
Education	High school	0.6% (2)	2.0% (19)
	Two-year college	3.1% (10)	6.5% (63)
	Four-year university	67.8% (219)	65.5% (631)
	Master’s degree	24.5% (79)	21.4% (206)
	Doctoral degree	4.0% (13)	4.7% (45)
Working	5 years or less	5.0% (16)	14.8% (143)
Year	6–10 years	12.4% (40)	17.2% (166)
	11–20 years	27.6% (89)	41.4% (399)
	21–30 years	52.9% (171)	22.2% (214)
	31 years or more	2.2% (7)	4.4% (42)
Position	Staff	3.7% (12)	10.8% (104)
	Senior staff	9.3% (30)	11.2% (108)
	Manager	30.3% (98)	23.5% (227)
	Deputy general manager	29.1% (94)	16.4% (158)
	General manager	6.2% (20)	27.6% (266)
	Team director	0.3% (1)	9.9% (95)
	Executive	0.0% (0)	0.6% (6)
Total		100% (323)	100% (964)

**Table 3 behavsci-12-00274-t003:** Definition and measurement of variables.

	Variable	Operational Definition	Measure	#Item
Independent variables	Agility	The degree to which an organization’s ability to respond effectively and quickly to changes in the market, supply, and demand in the development of competitive behavior and opportunities for innovation	Sambamurthy et al. [6]	3
Mediating variable	Helping behavior	The degree to which behaviors are intended to benefit others and the organization to which they belong	Fleishman [42]	4
	Knowledge sharing	The degree to which team members share work-related ideas, information, and suggestions with each other	Srivastava et al. [35]	4
Moderating variable	Customer orientation	The degree of effort to fully understand target customers and continuously create superior value for them	Saxe and Weitz [43]	3
Dependent variable	Innovativeness	Organizational openness to new ideas and experimental processes	Covin and Slevin [31]	3

**Table 4 behavsci-12-00274-t004:** Validity and reliability analysis result.

		Validity (Exploratory Factor Analysis)						Reliability	
		1 Knowledge Sharing	2 Helping Behavior	3 Customer Orientation	4 Innovativeness	5 Agility	Communality	Alpha When Item Is Deleted	Cronbach’s α
Knowledge sharing #2		0.830					0.833	0.896	0.924
Knowledge sharing #3		0.824					0.826	0.898	
Knowledge sharing #4		0.809					0.807	0.904	
Knowledge sharing #1		0.795					0.798	0.905	
Helping behavior #3			0.805				0.806	0.853	0.895
Helping behavior #4			0.767				0.757	0.881	
Helping behavior #2			0.764				0.777	0.860	
Helping behavior #1			0.750				0.761	0.868	
Customer orientation #1				0.812			0.782	0.767	0.842
Customer orientation #2				0.802			0.777	0.764	
Customer orientation #3				0.750			0.711	0.809	
Innovativeness #2					0.837		0.805	0.825	0.868
Innovativeness #1					0.804		0.808	0.795	
Innovativeness #3					0.786		0.775	0.821	
Agility #3						0.762	0.778	0.721	0.818
Agility #2						0.758	0.749	0.782	
Agility #1						0.641	0.709	0.749	
Eigenvalue		3.287	2.979	2.490	2.463	2.041			
Variance explained (%)		19.337	17.525	14.644	14.488	12.006			

**Table 5 behavsci-12-00274-t005:** Validity analysis result through confirmatory factor analysis.

χ^2^	*df*	CMIM/*df*	RMR	GFI	AGFI	CFI	TLI	NFI	IFI	RMSEA
304.566	109	2.794	0.016	0.971	0.960	0.987	0.983	0.980	0.987	0.037

**Table 6 behavsci-12-00274-t006:** Effect by path for ICT companies.

Path	β	SE	*t*	*p*
(Total effect of agility on innovativeness)				
Constant	0.000	0.044	0.000	1.000
Agility → Innovativeness (H1)	0.624 ***	0.044	14.298	0.000
(Direct effects on helping behavior)				
Constant	−0.032	0.051	−0.628	0.531
Agility → Helping behavior (H2)	0.346 ***	0.061	5.673	0.000
(Direct effects on knowledge sharing)				
Constant	0.000	0.039	0.000	1.000
Agility → Knowledge sharing	0.086	0.047	1.828	0.068
Helping behavior → Knowledge sharing (H3)	0.665 ***	0.047	14.233	0.000
(Direct effects on innovativeness)				
Constant	0.000	0.042	0.000	1.000
Agility → Innovativeness (H1”)	0.501 ***	0.051	9.857	0.000
Helping behavior → Innovativeness	0.076	0.065	1.182	0.238
Knowledge sharing → Innovativeness (H4)	0.179 **	0.061	2.954	0.003

** *p* < 0.01, *** *p* < 0.001.

**Table 7 behavsci-12-00274-t007:** Effect by path for non-ICT companies.

Path	β	SE	*t*	*p*
(Total effect of agility on innovativeness)				
Constant	0.000	0.025	0.000	1.000
Agility → Innovativeness (H1)	0.620 ***	0.025	24.488	0.000
(Direct effects on helping behavior)				
Constant	−0.040	0.031	−1.815	0.189
Agility → Helping behavior (H2)	0.322 ***	0.036	9.068	0.000
(Direct effects on knowledge sharing)				
Constant	0.000	0.023	0.000	1.000
Agility → Knowledge sharing	0.164 ***	0.027	6.154	0.000
Helping behavior → Knowledge sharing (H3)	0.598 ***	0.027	22.377	0.000
(Direct effects on innovativeness)				
Constant	0.000	0.024	0.000	1.000
Agility → Innovativeness (H1)	0.486 ***	0.029	17.044	0.000
Helping behavior → Innovativeness	0.114 ***	0.035	3.317	0.001
Knowledge sharing → Innovativeness (H4)	0.169 ***	0.034	4.999	0.000

*** *p* < 0.001.

**Table 8 behavsci-12-00274-t008:** Serial mediating effect.

Model							Sector	β	SE	BC 99% CI *
Agility	→	Helping behavior	→	Knowledge sharing	→	Innovativeness	ICT	0.041 **	0.018	0.0028 **iv
							Non-ICT	0.033 **	0.009	0.0129 **iv

* Bias-corrected 99% confidence interval. ** *p* < 0.01.

**Table 9 behavsci-12-00274-t009:** Moderating effect for ICT companies.

Path	β	SE	*t*	*p*
Constant	−0.032	0.051	−0.628	0.531
Agility → Helping behavior	0.346 ***	0.061	5.673	0.000
Customer orientation → Helping behavior	0.323 ***	0.063	5.172	0.000
Agility × customer orientation → Helping behavior	0.047	0.035	1.357	0.176

*** *p* < 0.001.

**Table 10 behavsci-12-00274-t010:** Moderating effect for non-ICT companies.

Path	β	SE	*t*	*p*
Constant	−0.040	0.031	−1.315	0.189
Agility → Helping behavior	0.322 ***	0.036	9.068	0.000
Customer orientation → Helping behavior	0.279 ***	0.036	7.740	0.000
Agility × customer orientation → Helping behavior	0.063 **	0.022	2.877	0.004

** *p* < 0.01, *** *p* < 0.001.

**Table 11 behavsci-12-00274-t011:** Effect by customer orientation.

Customer Orientation	β	SE	BC 99% CI *
−1SD	0.260 **	0.043	0.150–0.369
+1SD	0.385 **	0.041	0.280–0.490

* Bias-corrected 99% confidence interval. ** *p* < 0.01.

**Table 12 behavsci-12-00274-t012:** Moderated mediating effect according to customer orientation.

Customer Orientation	β	SE	BC 99% CI *
−1SD	0.026 **	0.008	0.008–0.052
+1SD	0.039 **	0.010	0.015–0.068

* Bias-corrected 99% confidence interval. ** *p* < 0.01.

## Data Availability

Not applicable.

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
