# Peer review of "Agility and Innovativeness: The Serial Mediating Role of Helping Behavior and Knowledge Sharing and Moderating Role of Customer Orientation"

_behavsci, 2022, doi:10.3390/bs12080274_

Round 1
Reviewer 1 Report
I consider the paper presented for review is very good, nevertheless, I suggest the Authors to add the following changes:
- Hypotheses should be more detailed and relate to the studied population, for example "Agility positively affects innovativeness." It is a very general formulation. A similar remark applies to the second hypothesis "Agility positively affects helping behaviors" and the next two hypotheses (3 and 4), the more so that the authors state "However, it is an overly simplistic claim that the same phenomenon occurs for all members under all circumstances.";
- I propose to add to the hypothesis the information that the research concerns customers of Information and Communication Technology (ICT) companies;
- The Abstract should include information on the size and form of selecting the research sample;
- I suggest a more precise title for Figure 2 and adding the information that it is own study;
- The information on the respondents is too cumulative in the text, moreover, all the data are in Table 2, I propose to rearrange the order (first the table, then the description), and mention only the main information in the description, as the text is illegible in its current form;
- How was the research sample selected?
- The results do not seem reliable as all respondents answered all the questions, which does not seem possible with a sample of more than 1,200 respondents. Therefore, information about the method of selecting the research sample is all the more necessary;
- The literature is relatively old, it would be advisable to refer to the new publications, at least in the chapter "Discussion".
Author Response
We would like to thank you for your valuable comments and suggestions. Based on the comments and suggestions, we supplemented and revised the paper as follows.
"I consider the paper presented for review is very good, nevertheless, I suggest the Authors to add the following changes:
- Hypotheses should be more detailed and relate to the studied population, for example "Agility positively affects innovativeness." It is a very general formulation. A similar remark applies to the second hypothesis "Agility positively affects helping behaviors" and the next two hypotheses (3 and 4), the more so that the authors state "However, it is an overly simplistic claim that the same phenomenon occurs for all members under all circumstances.";
I propose to add to the hypothesis the information that the research concerns customers of Information and Communication Technology (ICT) companies;"
-> As shown in the example below, as suggested by the reviewer, a more specific restrictive environment of ICT companies was included in all hypotheses.
- (before) Agility positively affects innovativeness.
- (after) Agility positively affects innovativeness in ICT companies
- The Abstract should include information on the size and form of selecting the research sample;
-> Information on the size and form of selecting the research sample was added to the abstract.
- How was the research sample selected?
- The results do not seem reliable as all respondents answered all the questions, which does not seem possible with a sample of more than 1,200 respondents. Therefore, information about the method of selecting the research sample is all the more necessary;
- The information on the respondents is too cumulative in the text, moreover, all the data are in Table 2, I propose to rearrange the order (first the table, then the description), and mention only the main information in the description, as the text is illegible in its current form;
-> Unnecessary and redundant information that could be identified in Table 2 were deleted, and instead, information on how the research sample was collected was described in more detail.
- I suggest a more precise title for Figure 2 and adding the information that it is own study;
-> As shown in the example below, the title of Table 2 was revised in more detail.
- (before) Research model
- (after) The working mechanism research model of the Agile approach
- The literature is relatively old, it would be advisable to refer to the new publications, at least in the chapter "Discussion".
-> In the Discussion chapter, some recently published references have been added.
Please let us know if you have any questions.
Thanks again to the reviewers for their nice comments and suggestions.

Reviewer 2 Report
Dear authors,
This research paper describes the actual topic – Agility and Innovativeness: The Serial Mediating Role of Helping Behavior and Knowledge Sharing and Moderating Role of Customer Orientation. In their article authors notice, that their study aims to understand the working mechanism of Agile approach by analyzing the effect of agility on innovativeness, sequential mediating effect of helping behavior and knowledge sharing, and moderating effect of customer orientation. As well, authors notice, that this study is meaningful in suggesting practical implications that it is important to create a culture that pursues “customer value” while promoting mutually helping behavior and sharing knowledge when introducing Agile methodology.
And I would like to share with authors some doubts and remarks too: it seems important to notice, that it would be needed to concentrate on the discussion and conclusions of the study. Thus, when developing these sections, the section of "Conclusions" or "Concluding insights" would be needed. Thus, it would be needed to include to the debate more future oriented theoretical implications, thus accessing deeper discussion and concluding insights.
Author Response
We would like to thank you for your valuable comments and suggestions. Based on the comments and suggestions, we supplemented and revised the paper.
As suggested, a “Conclusions” chapter was added to organize and concentrate important notices. Along with that, the limitations of this research were include to the debate more future oriented theoretical implications, thus accessing deeper discussion and concluding insights.
Please let us know if you have any questions.
Thanks again to the reviewers for their nice comments and suggestions.

Round 2
Reviewer 2 Report
Congratulations for your efforts to review the article. It's seems better now.